# Optimizing Axial and Peripheral Substitutions in Si-Centered Naphthalocyanine Dyes for Enhancing Aqueous Solubility and Photoacoustic Signal Intensity

**DOI:** 10.3390/ijms24032241

**Published:** 2023-01-23

**Authors:** Mohammad Ahsan Saad, Robert Pawle, Scott Selfridge, Leslie Contreras, Marvin Xavierselvan, Christopher D. Nguyen, Srivalleesha Mallidi, Tayyaba Hasan

**Affiliations:** 1Wellman Center for Photomedicine, Massachusetts General Hospital, Harvard Medical School, Boston, MA 02114, USA; 2Akita Innovations, North Billerica, MA 01862, USA; 3College of Engineering, Northeastern University, Boston, MA 02115, USA; 4Department of Biomedical Engineering, Science and Technology Center, Tufts University, Medford, MA 02155, USA; 5Division of Health Sciences and Technology, Harvard University and Massachusetts Institute of Technology, Cambridge, MA 02139, USA

**Keywords:** photoacoustic imaging, image-guided therapy, naphthalocyanine dyes, silicon-centered naphthalocyanine dyes, water-soluble naphthalocyanines

## Abstract

Photoacoustic imaging using external contrast agents is emerging as a powerful modality for real-time molecular imaging of deep-seated tumors. There are several chromophores, such as indocyanine green and IRDye800, that can potentially be used for photoacoustic imaging; however, their use is limited due to several drawbacks, particularly photostability. There is, therefore, an urgent need to design agents to enhance contrast in photoacoustic imaging. Naphthalocyanine dyes have been demonstrated for their use as photoacoustic contrast agents; however, their low solubility in aqueous solvents and high aggregation propensity limit their application. In this study, we report the synthesis and characterization of silicon-centered naphthalocyanine dyes with high aqueous solubility and near infra-red (NIR) absorption in the range of 850–920 nm which make them ideal candidates for photoacoustic imaging. A series of Silicon-centered naphthalocyanine dyes were developed with varying axial and peripheral substitutions, all in an attempt to enhance their aqueous solubility and improve photophysical properties. We demonstrate that axial incorporation of charged ammonium mesylate group enhances water solubility. Moreover, the incorporation of peripheral 2-methoxyethoxy groups at the α-position modulates the electronic properties by altering the π-electron delocalization and enhancing photoacoustic signal amplitude. In addition, all the dyes were synthesized to incorporate an N-hydroxysuccinimidyl group to enable further bioconjugation. In summary, we report the synthesis of water-soluble silicon-centered naphthalocyanine dyes with a high photoacoustic signal amplitude that can potentially be used as contrast agents for molecular photoacoustic imaging.

## 1. Introduction

Photoacoustic (PA) imaging is based on the excitation of certain molecules by a nanosecond pulsed beam of light resulting in thermal expansion and subsequent contraction of the surrounding media, followed by propagation of pressure waves detected through ultrasound transducers [1,2]. Contrast-enhanced PA imaging has been established as a potential modality to provide improved sensitivity and resolution [3,4]. While the more routinely used imaging modalities such as X-ray, computed tomography (CT), and positron emission tomography (PET) provide high imaging resolution and sensitivity, the use of ionizing radiation, toxicity of contrast agents, longer data acquisition time, and limited real-time imaging capability are major drawbacks [5]. In contrast, PA imaging can provide high spatial resolution with excellent depth profiling, uses non-ionizing radiation, and can be used in real-time, thus providing an option with combined benefits of ultrasound and fluorescence imaging [4,6,7].

There are several endogenous biomolecules that can enhance PA contrast. These include collagen, oxygenated and deoxygenated hemoglobin (blood), and lipids [4,8]. Importantly, PA imaging is approved for clinical use in breast cancer diagnosis and is in early-phase human trials for many other cancers, including skin, head, and neck cancers (NCT03897270, NCT04110249) [3,9,10]. In order to further improve the contrast in PA imaging, externally administered contrast agents can be added [11]. Many synthetic PA contrast agents have been reported, including organic compounds such as porphyrin and cyanine-based dyes and several metallic and organic nanoparticles [12,13]. As electromagnetic radiation absorbed by molecules dissipates through a combination of radiative and non-radiative decay, several external contrast agents, which are currently approved by the FDA or those in clinical trials for fluorescence imaging, such as Indocyanine Green (ICG), methylene blue (MB) and IRDye800, have also been used pre-clinically as PA contrast agents [14,15,16,17,18]. While their quantum yield of non-radiative decay is limited, the PA contrast enhancement by these dyes can still provide an advantage over endogenous sonochromes.

Although IRDye800 and ICG can be used as PA contrast agents, these dyes suffer from one of several drawbacks, including low photostability, unstable pharmacokinetics, aggregation, and variability in PA signal intensity depending on the medium and dye concentration. Moreover, the significantly high quantum yield of fluorescence of these dyes limits their non-radiative decay quantum yields, and their spectral overlap with endogenous chromophores, especially hemoglobin, makes it difficult to extract and separate the PA signal of the externally administered dye from the tissue background [13,19,20,21,22]. ICG has also been reported to have issues with conjugation to targeting moieties [13].

In order to circumvent the aforementioned issues, there have been continuous attempts to develop dyes for potential use as contrast agents for PA imaging. Si-centered and other metal-centered naphthalocyanine dyes have been reported previously for their use as PA contrast imaging agents [23,24,25]. We have also reported the synthesis, characterization, and conjugation of a Si-centered naphthalocyanine (SiNC) dye for molecular-targeted PA imaging [25,26]. However, a shortcoming of these SiNC dyes is their limited water solubility. In this study, we build up on our previous work by developing silicon-centered naphthalocyanine dyes with increasing aqueous solubility while retaining their photophysical properties. Moreover, these dyes are designed to incorporate an N-hydroxysuccinimidyl group for spontaneous bond formation with an amine group and bioconjugation to target moieties.

## 2. Results

### 2.1. Synthesis of Silicon Naphthalocyanine Dyes SiNC(1–4)

Naphthalocyanines have several substituent positions. There are 24 potential substituent sites on the naphthalocyanine macrocycle in three different locations labeled α, β, γ in Figure 1. The central element may contain additional substituent sites depending on the specific element. Figure 1 shows the structure of naphthalocyanine dyes SiNC(1–4). All dyes are silicon-centered naphthalocyanines with eight alkoxy substituents, all in the α-positions of the macrocycle. This basic structure yields the near-infrared absorbance of the dye at approximately 864 nm. Using silicon as the central element adds two extra bonding sites to the naphthalocyanine via silicon-oxygen-silicon bonds, which are hydrolytically stable in biological systems. Each dye has an N-succinimidyl (NHS) ester (denoted as R1 in Figure 1) in one of the silicon substituent positions, bound via siloxy and thioether linkages. The NHS ester provides bioconjugation groups and ensures that each dye molecule can form only a single bioconjugate bond. The other bonding site on the silicon center is a hydroxyl group in SiNC-1, charged ammonium mesylate group in SiNC-2 and SiNC-4, and charged ammonium mesylate and ethylene glycol groups in SiNC-3 (denoted as R2 in Figure 1). With regards to peripheral substitution, SiNC(1–3) have butoxy groups in the alpha positions on the macrocycle, while SiNC-4 has methoxyethoxy groups in the alpha positions.

SiNC(1–4) were prepared according to Figure 1. The preparation of silicon naphthalocyanine dihydroxides substituted with butoxy groups (5) or methoxyethoxy groups (6) is discussed in the Materials and Methods section. Thiol groups were incorporated on one or both of the central silicon bonding sites via siloxy linkages, yielding 7 from 5, 8 from 5, and 9 from 6. 7 and 8 were obtained from a single reaction by adjusting the ratio of starting materials. NHS esters were incorporated via a Michael addition of the thiols to NHS acrylate, yielding 1 from 7, 10 from 8, and 11 from 9. Finally, SiNC(2–4) dyes were prepared by exchanging one of the NHS groups on 10 or 11 with bis(dimethylpropylamino)amine followed by amine quaternization with methyl mesylate, yielding 2 from 10 and 4 from 11, or triethylene glycol monomethyl ether methanesulfonate (preparation discussed in the Materials and Methods section), yielding 3 from 10. The physicochemical characteristics of the four dyes are summarized in Table 1.

### 2.2. Photophysical Characterization of SiNC(1–4) Dyes

The absorption spectra of SiNC(1–4) are shown in Figure 2. The UV-Vis spectra show a typical Q-band characteristic of a substituted macrocycle ring around 850 nm. A vibrational shoulder around 750 nm was also observed for all the SiNC(1–4) dyes and is in agreement with previously reported data for similar dyes [27]. For SiNC-1 and SiNC-2, dispersion in aqueous solvents was not possible, but SiNC-2 was dispersible by adding a small amount of ethanol to the solvent system. For SiNC-4, a decrease in the ratio between the absorbance of the Q-band and the vibronic shoulder was observed in water. This is characteristic of H-aggregation. In contrast, a red-shifted peak characteristic of J-aggregate formation [28,29,30] and lateral coupling was observed for SiNC-3 (Figure 2C) in water. This red shift in the peak was not observed for SiNC-2 and SiNC-4. The ratio between the Q-band and the vibrational shoulder, as calculated by the normalized absorption spectra in the denoted solvent, was found to be higher for SiNC-2 and SiNC-3 (~3.3 and 4.0 for SiNC-2 and SiNC-3, respectively), suggesting minimal H-aggregation. However, for SiNC-4, the ratio was ~2.5 (in water, represented by the blue dotted line in Figure 2D), suggesting relatively higher H-aggregation.

### 2.3. Photophysical Characterization of SiNC(1–4) Dyes in Buffer

While the absorption profiles shown in Figure 2 can be predictive of SiNC(1–4) dye absorption properties in other solvents, monitoring absorption in aqueous buffers and other biologically relevant media is important to determine the exact performance of these dyes in a biological environment. PA signal amplitude is dependent on the optical absorption properties of the dye being imaged, including their extinction coefficients and quantum yield of non-radiative decay. To compare the absorption properties of the SiNC(1–4) dyes, we recorded their absorption spectra in PBS and compared them with that of the standard clinically approved dyes, ICG and IRDye800. While the solubility of these dyes differs, we preferred PBS and PBS with BSA to mimic physiological conditions. As shown in Figure 3, the absorption spectra of all the SiNC(1–4) dyes, except SiNC-1, showed a peak in the range of 850–930 nm. SiNC-1 did not show any absorption peak, possibly due to aggregation in the aqueous buffer, and was found to be insoluble even in the presence of 0.1% BSA at the dye concentration (5 µM) that we tested. For SiNC-2 and SiNC-3, dispersion in PBS and PBS-BSA resulted in the shift of the absorption maxima towards the red region beyond 900 nm, suggesting J-aggregate formation. In our previous study, we have shown the influence of BSA on the absorption and PA properties of a similar SiNC dye [25]. We, therefore, studied the effect of BSA on the absorption profile of SiNC(1–4), ICG, and IRDye800. The addition of BSA did not influence the absorption properties significantly. For all the SiNC dyes, the addition of 0.1% BSA increased the absorption for SiNC-2 and SiNC-3 at 935 nm and 930 nm, respectively. The addition of 0.1% BSA also increased the absorption for ICG with a slight red-shift in the absorption maxima to 789 nm. The absorbance of IRDye800 was essentially unchanged.

Figure 4A shows the PA spectra (680 nm to 970 nm) of the dyes with and without BSA (acquired as described in Appendix A). The PA spectra for all the dyes appeared to be similar to their absorption spectra shown in Figure 3. The PA signal intensity at 880 nm (for SiNC(1–4) dyes) and 780 nm (for ICG and IRDye800) (Figure 4B), respectively, suggested that the SiNC(1–4) dyes and ICG had a relatively higher PA signal intensity as compared to IRDye800, with SiNC-4 showing the highest PA signal. However, with the addition of BSA, the PA signal intensity of SiNC-4 and IRDye800 decreased significantly. The decrease in PA signal intensity of ICG after the addition of BSA was not significant. Interestingly, the addition of BSA led to an increase in the PA signal intensity of SiNC-1, SiNC-2, and Si-NC3. The PA signal intensity of SiNC(1–3) in BSA was 3–9 folds higher than that for SiNC-4 in BSA.

### 2.4. Photobleaching Studies

As mentioned previously, the PA signal amplitude of dyes depends on their optical absorption properties. Many dyes undergo photobleaching upon laser irradiation. Continuous monitoring of change in PA signal amplitude is indicative of photobleaching of the dye, i.e., the larger the decrease in PA signal amplitude implies greater photobleaching of the dye during the imaging session. Pulse-to-pulse PA response of ICG in water is shown as an example in Figure 5A and is representative of the measurements obtained with our custom-built PA imaging system, as reported previously [31]. Here we observe that the PA signal has indeed decreased with an increased number of laser pulses exposed to the dye. A change in PA signal over 3000 nanosecond laser pulses of fluence of 30 mJ/cm^2^ is plotted in Figure 5B. A change in the PA signal is indicative of the photobleaching of the dyes. ICG in water showed the largest change in PA signal over time (Figure 5B), and this is indicative of its poor stability upon light irradiation and correlates well with our previous report [32]. On the contrary, ICG in DMSO was stable and did not photobleach. IRDye800 had a similar level of stability in both water and DMSO. The NC dyes in DMSO were highly stable when compared to ICG in water. Amongst the SiNC(1–4) dyes, SiNC-4 had the lowest change in PA signal, indicative of less photobleaching effects due to light irradiation. Furthermore, a statistically significant difference was observed between ICG and SiNC-4 dye. When utilizing dyes for PA imaging, it is imperative for the signal to remain unchanged during the imaging session. As SiNC-4 has the highest photostability amongst the dyes, it is an apt PA imaging contrast agent.

### 2.5. Cellular Uptake and PA Imaging

To further evaluate the utility of these dyes for possible application as PA contrast agents, we incubated the SiNC(1–4) dyes, ICG, and IRDye800 with Cal 27 cells for 2 h and 24 h. The human tongue squamous cell carcinoma line (Cal 27) was used to test the uptake and imaging efficacy of the dyes in an in vitro oral cancer model. After the desired incubation periods, 0.25 × 10^6^ cells were collected, and tumor phantoms were prepared. The phantoms were imaged at 780 nm and 880 nm, and the signal intensity was analyzed and plotted as shown in Figure 6. Figure 6A shows the experimental scheme, and Figure 6B shows the PA images and the corresponding ultrasound images of tumor phantoms prepared from Cal 27 cells treated with different dyes for different durations. While phantoms prepared with gelatin and untreated cells did not show any signals (Figure 6B), either at 780 nm or 880 nm, cells treated with the various SiNC(1–4) dyes showed signals with varying intensities (Figure 7). A high PA signal intensity was obtained for SiNC-2, SiNC-3, and SiNC-4, which increased with time. While the PA signal for SiNC-1 was detectable at the 2-h time-point, the signal intensity diminished after 24 h of incubation. For ICG and IRDye800, although the signal intensity increased with time, it was significantly lower as compared to the SiNC(1–4) dyes. Although previous studies have shown the uptake of ICG and IRDye800 in different cell lines [33], the relatively lower PA contrast for the ICG and IRDye800 treated phantoms obtained in the present study was possibly due to the lower cell numbers in the tumor phantoms and the acquisition settings used. The acquisition settings were chosen to prevent over-saturation of the PA signal of SiNC(2–4) treated phantoms. The PA signal with amplified contrast for ICG, and IRDye800 treated phantoms is provided in Appendix A.

## 3. Discussion

In this study, we report the effect of peripheral and axial substitutions on the naphthalocyanine macrocycle to synthesize a water-soluble naphthalocyanine dye for potential application as a PA contrast agent. PA imaging can be potentially transformative in the diagnosis, treatment guidance, and monitoring of cancerous and non-cancerous pathologies [34,35,36]. While agents such as ICG [37,38] and IRDye800CW [39], which have been reported for clinical use as fluorophores, can be used as PA contrast agents, their use in PA contrast enhancement is limited due to their low quantum yield of non-radiative decay, rapid excretion and short circulation time, and concentration-dependent aggregation and changes in absorption profiles [13,19,20,21,22]. Moreover, their absorption profile overlaps with that of hemoglobin, making it difficult to delineate the signal from the background [40]. In contrast, naphthalocyanine dyes have been reported to have a significantly higher PA contrast due to high extinction coefficients and a high non-radiative quantum yield [17,18] while also being reported for photosensitization therapy [41]. Their electronic properties can be tuned by the substitution of several (24 positions) available peripheral groups, changing the metal atom in the inner porphyrazine ring or through axial modifications to the central metal atom [27]. Substitution of metal atoms in the central position alters the extinction coefficients of these molecules while also affecting solubility [18]. Further conjugating the metal atom with chemical moieties, such as polyethylene glycol (PEG), can also increase solubility and alter electronic properties [18,23,42]. It has been reported previously that substitution of functional groups to the central metal atom or to the inner ring (α-positions) can tune the absorption wavelengths in the NIR region, while substitutions at the outer rings affect the absorption intensities without influencing the absorption wavelength [43]. The presence of a silicon center and octabutoxy groups on the alpha position alter the absorption profile of these dyes resulting in a higher extinction coefficient at a red-shifted absorption wavelength [24,25,44]. An important aspect that needs to be modulated in the context of the clinical use of naphthalocyanine dyes is their aqueous solubility. Most naphthalocyanine dyes are hydrophobic with a tendency to aggregate due to the strong pi-pi interactions between the planar structure [45].

While the axial conjugation of polyethylene glycol has been reported previously to increase the solubility of naphthalocyanine dyes [23,42], silicon-oxygen-carbon bonds are not stable to hydrolysis and characterization of molecules using PEG as the solubilizing group is difficult. Alternate solubilizing groups that do not have these issues are highly desired. Silicon-oxygen-silicon bonds resist hydrolysis. Charged groups or short, length-specific oligoethylene glycols (di-, tri-, and tetra-ethylene glycols) allow for explicit characterization via NMR and mass spectrometry. Following these design parameters, we investigated axial substitutions with ammonium mesylate and diethylene glycol groups bound to the central silicon via hydrolytically stable silicon-oxygen-silicon linkages. We used octabutoxy and methoxyethoxy groups as peripheral substituents, hypothesizing that the methoxyethoxy groups would improve water solubility while maintaining the absorbance wavelength of octabutoxy-substituted naphthalocyanines. In previous studies with Si-naphthalocyanine dye, axial and peripheral substitutions are reported; however, dyes with both peripheral and axial substitutions have not been reported. To the best of our knowledge, this is the first study combining peripheral and axial substitutions in a Si-centered NC dye. In addition, we also maintain an N-succinimidyl (NHS) ester, bound via siloxy and thioether linkages to the central Si atom, which can assist in bioconjugation to ligands for targeting the NC dye. Previous reports on PEGylated Si-centered naphthalocyanine dyes reported accumulation of the dye in healthy tissue along the target tumor site, limiting their targeting ability [42], thus highlighting the need for targeting these molecules. In our previous study, we have shown the bioconjugation of a Si-centered naphthalocyanine dye, with peripheral octabutoxy substitutions, to an anti-Epidermal growth factor receptor (EGFR; cetuximab) antibody for targeting over-expressed EGFR in tumor cells, which can potentially be used for PA imaging in deep-seated tumors [25,26].

In the present study, we were able to develop Si-centered naphthalocyanine derivatives with high water solubility. SiNC-2, SiNC-3, and Si-NC-4 had an axial substitution of charged ammonium mesylate group and showed high PA intensity, with SiNC-4 showing the highest intensity. The high PA intensity for SiNC-4 could possibly be due to the oxygen atoms that could enable solvent molecules to come closer to the ring, encouraging non-radiative decay via electron transfer, although its absorption in PBS was similar to that of SiNC-3 but lower than that of SiNC-2. In terms of cellular uptake, SiNC-3 showed the highest intensity, which may be attributed to an enhanced cellular uptake due to the diethylene glycol moieties in the axial substituent. In our comparative studies, the absorption of SiNC-1 was found to be the lowest in PBS due to aggregation, and the PA intensity was the highest, which again is due to the aggregation of these molecules. The incorporation of solubilizing groups into the silicon center increased the solubility (as evidenced by a distinct absorption peak); however, the PA intensity decreased due to a reduction in J aggregate formation. BSA had minimal effect in lowering the PA intensity for SiNC-2 and SiNC-3. However, for SiNC-4, BSA decreased the PA signal intensity, which was also the case with IRDye800 and ICG, suggesting that water-soluble dyes had more impact from the surrounding biomolecules than their hydrophobic counterparts. This may be due to the enhanced interaction of the water-soluble dyes with the amphiphilic BSA molecules, thus inhibiting any supramolecular arrangement (J-aggregates), which is usually suggested to enhance the PA properties of these dyes. Solubilization of the dyes by substitution and incorporation of solubilizing groups also had a significant impact on the cellular uptake of these dyes. While the uptake of NC-1 appeared to be rapid, its signal decreased over a period of 24 h. However, for SiNC(2–4), the PA signal from cellular phantoms increased till 24 h, suggesting rapid and sustained accumulation, probably due to the amphiphilic nature of these dyes. For ICG and IRDye800CW, the signal was low, which, although increased over time, was still significantly lower than the SiNC(2–4). This could be due to the lower cellular uptake, low extinction coefficients, and the low quantum yield of non-radiative decay.

## 4. Materials and Methods

### 4.1. Synthesis of Silicon Naphthalocyanine Dyes

#### 4.1.1. Preparation of Compound 1

We have previously discussed the preparation of Compound 5 [25]. For the synthesis of Silicon 5,9,14,18,23,27,32,36-octa(butoxy)-2,3-naphthalocyanine [(3″-mercaptopropyl)dimethylsiloxide] monohydroxide (7) and silicon 5,9,14,18,23,27,32,36-octa(butoxy)-2,3-naphthalocyanine di[(3″-mercaptopropyl)dimethylsiloxide] (8), a 100 mL single-neck round bottom flask was charged with a stir bar, 480 mg (0.355 mmol) silicon 5,9,14,18,23,27,32,36-octa(butoxy)-2,3-naphthalocyanine dihydroxide (5), 0.8 mL (790 mg, 4.8 mmol) methoxydimethylsilylpropanethiol. The flask was fitted with a short path distillation setup, and the vessel was purged with argon and evacuated and refilled with argon twice. The vessel was wrapped with foil and immersed in a 140 °C oil bath, and the mixture was stirred. When pyridine began to reflux, the oil bath temperature was turned down to 130 °C, and the mixture was refluxed for 5.5 h. The heat was turned off, and the mixture stood at room temperature for 60 h. The solvent was then removed in vacuo. A total of 25 mL of toluene was added to the mixture, and the solvent was removed in vacuo. The residue was dissolved in 5 mL dichloromethane and precipitated into 90 mL methanol. The precipitate was isolated by centrifugation and purified by CombiFlash chromatography on a 24 g pre-packed silica column with gradient elution, 3:1 hexane:dichloromethane to 1:1 hexane:dichloromethane to dichloromethane as eluent. Two fractions were collected, one with 3:1 hexane:dichloromethane eluent and one with dichloromethane eluent. The solvents were removed from the first collected fraction in vacuo, yielding 192 mg (33.5%) 8. The solvents were removed from the second collected fraction in vacuo, yielding 196 mg (37%) 7. Both compounds are brown crystalline solids. A total of 1H NMR (500 MHz, CDCl3) of 7: δ = 9.02 (m, 8H), 7.94 (m, 8H),5.22 (t, J = 9 Hz, 16H), 2.27 (quin, J = 9 Hz, 16H), 1.67 (sextet, J = 10 Hz, 16H), 1.21 (q, J = 9 Hz, 2H), 1.06 (t, J = 9 Hz, 24H), 0.15 (m, 1H), −0.42 (m, 2H), −1.51 (m, 2H), −2.19 (s, 6H). A total of 1H NMR (500 MHz, CDCl3) of 8: δ = 9.00 (m, 8H), 7.91 (m, 8H), 5.19 (t, J = 9 Hz, 16H), 2.25 (quin, J = 9 Hz, 16H), 1.66 (sextet, J = 10 Hz, 16H), 1.20 (q, J = 9 Hz, 4H), 1.03(t, J = 9 Hz, 24H), 0.15 (t, J = 10 Hz 2H), −0.43 (m, 4H), −1.53 (m, 4H), −2.21 (s, 12H).

Synthesis of Silicon 5,9,14,18,23,27,32,36-octa(butoxy)-2,3-naphthalocyanine mono[([([(N-succinimidyl)oxy]carbonyl)ethy]thio)propyldimethylsiloxide] monohydroxide (1): A 100 mL single-neck round bottom flask was charged with a stir bar, 186 mg (0.125 mmol) 7, and 29 mg (0.17 mmol) N-succinimidyl acrylate. The vessel was purged with argon. A solution of 0.3 mL diisopropylethylamine and 20 mL dichloromethane was degassed with argon and injected into the reaction vessel. The reaction vessel was immersed in a 40 °C oil bath, and the mixture was stirred 16 h. The solvents were removed in vacuo, and 25 mL of methanol was added to the mixture. The mixture was agitated with stirring, scraping, and sonication. The solids were isolated via centrifugation, and the solvent was removed in vacuo, yielding 168 mg (81%) 1 as a brown crystalline solid. UV/Vis (toluene): λmax = 873 nm, ε = 2.77 × 10^5^ M^−1^ cm^−1^ 1H NMR (500 MHz, CDCl3): δ = 9.02 (m, 8H), 7.94 (m, 8H),5.22 (t, J = 9 Hz, 16H), 2.27 (quin, J = 9 Hz, 16H), 2.11 (t, J = 10 Hz, 2H), 1.99 (t, J = 10 Hz, 2H), 1.68 (sextet, J = 10 Hz, 16H), 1.06 (t, J = 9 Hz, 24H), −0.42 (m, 2H), −1.53 (m, 2H), −2.19 (s, 6H).

#### 4.1.2. Preparation of Compound 2

Silicon 5,9,14,18,23,27,32,36-octa(butoxy)-2,3-naphthalocyanine di[([([(N-succinimidyl)oxy]carbonyl)ethy]thio)propyldimethylsiloxide] (10): A 100 mL three-neck round bottom flask was charged a stir bar, equipped with a Vigreux column, and evacuated and refilled with argon three times. The glassware and stir bar were dried in an oven and assembled prior to cooling. A total of 350 mg (0.217 mmol) 8 and 210 mg (1.24 mmol) N-succinimidyl acrylate were added to the reaction vessel. A solution of 30 mL dichloromethane and 0.75 mL diisopropylethyl amine was prepared, degassed with argon, and added to the reaction mixture. The reaction vessel was immersed in a 40 °C oil bath and stirred for 16 h. The solvent was removed in vacuo, and the crude product was washed with 25 mL methanol two times. The crude product was purified by CombiFlash chromatography on a 12 g pre-packed silica column, 5% ethyl acetate in dichloromethane eluent. The solvent was removed in vacuo, yielding 306 mg (72%) 10 as a brown crystalline solid. A total of 1H NMR (500 MHz, CDCl3): δ = 5.22 (br. s, 16H), 2.77 (s, 8H), 2.32 (br. s, 16H), 2.15 (t, J = 9 Hz, 4H), 2.01 (t, J = 9 Hz, 4H), 1.71 (sextet, J = 9 Hz, 16H), 1.06 (t, J = 9 Hz, 24H), −0.38 (m, 4H), −1.50 (m, 4H), −2.16 (s, 12H).

Silicon 5,9,14,18,23,27,32,36-octa(butoxy)-2,3-naphthalocyanine mono[([([(N-succinimidyl)oxy]carbonyl)ethy]thio)propyldimethylsiloxide] mono([([([bis(propyltrimethylammonium)amino]carbonyl)ethy]thio)propyl]dimethylsiloxide) dimesylate (2): A 20 mL scintillation vial was charged with 73 mg (0.037 mmol) 10 and a stir bar and capped with a septum and purged with argon. A total of 5 mL chloroform was added to the reaction vessel. A total of 0.1 mL (0.026 mol) of a 0.26 M solution of bis(dimethylaminopropyl)amine was added to the reaction mixture and the mixture stirred for 16 h at room temperature. A total of 40 µL of methyl methanesulfonate was added to the reaction mixture, and the mixture was stirred 16 h. The solvents were removed in vacuo, and the residue was washed with 30 mL diethyl ether. The crude product was centrifuged, and the supernatant was drained. The product was washed with an additional 20 mL of diethyl ether and centrifuged. The supernatant was drained, and the remaining solvent was removed in vacuo, yielding 27 mg (33%) 2 as a waxy brown solid. UV/Vis (chloroform): λmax = 873 nm, ε = 1.94 × 10^5^ M^−1^ cm^−1^ 1H NMR (500 MHz, CDCl3): δ = 5.23 (br. s, 16H), 3.52–3.10 (m, 30H), 2.64 (s, 4H), 2.61 (s, 6H), 2.37–2.30 (m, 16H), 2.05 (m, 8H), 1.71 (m, 16H), 1.39 (m, 2H), 1.35 (m, 2H), 1.05 (t, J = 7 Hz, 24H), −0.26 (m, 2H), −0.38 (m, 2H), −1.51 (m, 4H), −2.14 (s, 12H).

#### 4.1.3. Preparation of Compound 3

2,(2-(2-methoxyethoxy))ethoxy)ethoxy mesylate (14): A 100 mL round bottom flask was charged with a stir bar and flame dried with argon purge. The vessel was cooled, and 4 mL (4.19 g, 26 mmol) of triethylene glycol monomethyl ether and 50 mL dichloromethane were added to the vessel. The vessel was immersed in an ice bath for 15 min. A total of 6 mL (4.38 g, 43 mmol) triethyl amine was added to the reaction mixture, and the mixture was cooled in an ice bath for 5 min. A total of 3 mL (4.5 g, 39 mmol) methanesulfonyl chloride was added to the mixture dropwise over 5 min. The mixture became cloudy and was stirred 15 min in an ice bath. The mixture was removed from the ice bath and stirred an additional 15 min. The reaction mixture was washed with deionized water. The organic layer was removed from the reaction vessel via syringe transfer, filtered, and dried with sodium sulfate. The solvent was removed in vacuo, yielding 6.38 g (quantitative yield) 14 as a clear yellow oil. Analyses agree with the same product reported in the literature.

Silicon 5,9,14,18,23,27,32,36-octa(butoxy)-2,3-naphthalocyanine mono[([([(N-succinimidyl)oxy]carbonyl)ethy]thio)propyldimethylsiloxide] mono([([([bis([2,(2-[2-methoxyethoxy]ethoxy)ethyl]propyldimethylammonium)amino]carbonyl)ethy]thio)propyl]dimethylsiloxide) dimesylate (3): A single neck 100 mL round bottom flask was charged with 178 mg (0.091 mmol) 10 and a stir bar. The vessel was fitted with a septum and purged with argon. A total of 15 mL chloroform was injected into the vessel while argon continued purging. A total of 15 mg (0.080 mmol) bis(dimethylaminopropyl)amine was added to the mixture, and the mixture was stirred for 16 h at room temperature. Solvent was removed in vacuo until approximately 5 mL of mixture remained, then an additional 2.7 mg (0.014 mmol) bis(dimethylaminopropyl)amine was added to the mixture, and the mixture was stirred one additional hour. A total of 1.139 g (4.7 mmol) 14 was dissolved in 4 mL chloroform, dried with 3 A molecular sieves, and then centrifuged and filtered through a 1 um filter. A total of 1.5 mL of this solution was added to the reaction mixture, and the mixture was stirred 72 h at room temperature. The solvent was reduced in vacuo, and then the mixture was precipitated with diethyl ether. The mixture was centrifuged, the supernatant was decanted, and the precipitate was washed with diethyl ether. The centrifugation, decanting, and washing step was performed an additional time; then, the solvent was removed in vacuo, yielding 82 mg (36%) 3 as a waxy brown solid. UV/Vis: (solvent, λmax, ε): chloroform, 873 nm, 1.75 × 10^5^ M^−1^ cm^−1^; water, 934 nm, 0.93 × 10^5^ M^−1^ cm^−1^; 5% triton X-100 in water, 870 nm, 2.20 × 10^5^ M^−1^ cm^−1^. Δ = 1H NMR (500 MHz, C6D6): δ = 9.20 (m, 8H), 7.89 (br. s, 5H), 7.71 (br. s, 3H), 5.46 (br. s, 16H), 3.74–2.94 (m, 50H), 2.37 (m, 16H), 1.94–1.88 (m, 4H), 1.67 (m, 16H), 1.04 (t, J = 7 Hz, 24H), −0.19–0.22 (m, 4H), −1.30 (m, 4H), −1.85–1.89 (m, 12H). A total of 1H NMR (500 MHz, CDCl3): δ = 5.24–5.17 (br. s, 16H), 3.71–3.09 (m, 26H), 2.99–2.80 (m, 28H), 2.75 (s, 4H), 2.66–2.57 (m, 4H), 2.17 (m, 4H), 2.05 (m, 16H), 1.70 (m, 16H), 1.48–1.37 (m, 2H), 1.06 (t, J = 7 Hz, 24H), −0.12–0.22 (m, 4H), −1.31–1.52 (m, 4H), −1.95–2.16 (s, 12H).

#### 4.1.4. Preparation of Compound 4

2,3-dicyano-1,4-di(2′-methoxyethoxy)naphthalene (12): A 200 mL round bottom flask was charged with 4.95 g (23.6 mmol) 2,3-dicyano-1,4-dihydroxynaphthalene, 19.4 g (140 mmol) of freshly ground potassium carbonate, and a stir bar. The reaction vessel was sealed with a rubber septum. A total of 50 mL N,N-dimethylformamide, and 8 mL (11.86 g, 85.3 mmol) 2-bromoethyl methyl ether was added to the reaction mixture, and the headspace of the reaction vessel was purged with argon. The reaction vessel was placed in an oil bath set to 80 °C, stirred, and the vessel headspace was purged with argon. The oil bath temperature was increased to 100 °C, and the reaction mixture was stirred for 16 h. A total of 150 mL of water was added to the reaction mixture, and the mixture was stirred vigorously for five minutes. The mixture was filtered, with the solid filtrate washed with water (2 × 100 mL). Residual solvent was removed from the solid filtrate in vacuo, leaving a brown crude product. The crude product was purified via CombiFlash chromatography on a 22 g pre-packed silica column with gradient elution, dichloromethane to 1:1 dichloromethane: ethyl acetate as eluent. The relevant fractions were collected, combined, and the solvent was removed in vacuo, yielding 5.523 g (72%) of 2 as an off-white crystalline powder. A total of 1H NMR (400 MHz, CDCl3): δ = 8.37 (m, 2H), 7.79 (m, 2H), 4.56 (m, 4H), 3.85 (m, 4H), 3.47 (s, 6H). A total of 13C NMR (125 MHz, CDCl3) δ = 157.55, 130.71, 130.36, 123.88, 114.28, 99.60, 75.18, 71.46, 59.10. HR-ESI-MS: *m*/*z*; [M + H]^+^, calculated for C18H19O4N2, 327.1339; found: 327.1340.

5,9,14,18,23,27,32,36-octa(2′-methoxyethoxy)-2,3-naphthalocyanine (13): A 100 mL 3 neck round bottom flask was charged with 705 mg sodium hydride and a stir bar. The reaction vessel was evacuated and refilled with argon three times, and then the sodium hydride was washed with hexane (2 × 5 mL). The reaction vessel was evacuated and refilled with argon two times to remove the hexane. A total of 25 mL 2-methoxyethanol was added to the reaction vessel, resulting in significant gas generation. The reaction vessel was evacuated and refilled with argon three times, and then 2.70 g (8.28 mmol) of 2 was added to the reaction vessel. The reaction vessel was immersed in a 130 °C oil bath and stirred for 16 h. The reaction mixture was cooled to room temperature, and then 1 mL glacial acetic acid was added to the mixture. Half of the solvent (approximately 12 mL) was removed by distillation, and then the crude product was precipitated by adding 250 mL methanol. The crude product was removed by filtration, and then residual solvent was removed in vacuo. A total of 1.15 g (42.6%) of 6 was recovered as a brown crystalline solid. UV/Vis (solvent, λmax, ε): toluene, 858.1 nm, 2.47 × 10^5^ M^−1^ cm^−1^. A total of 1H NMR (400 MHz, C6D6): δ = 9.53 (m, 8H), 7.74 (m, 8H), 5.63 (m, 16H), 4.02 (m, 16H), 3.34 (s, 24H).

Silicon 5,9,14,18,23,27,32,36-octa(2′-methoxyethoxy)-2,3-naphthalocyanine dihydroxide (6): A 250 mL 3-neck round bottom flask, Vigreux column, and stir bar were oven dried, assembled while still hot, and evacuated and refilled with argon three times. The reaction vessel was charged with 110 mL dichloromethane, 10 mL triethylamine, and 1.13 g (0.865 mmol) 3. The reaction mixture was degassed with argon, and then 2.4 mL (24 mmol) trichlorosilane was added to the mixture. The reaction mixture turned a reddish-purple, and the mixture was stirred for 16 h at room temperature under argon. UV/Vis of an aliquot of the reaction mixture showed λmax = 930 nm, indicating SiCl_2_ insertion into the naphthalocyanine cavity. The reaction vessel was immersed in a 40 °C water bath, and the mixture was sparged with argon and vented through a water trap, evolving white smoke due to trichlorosilane hydrolysis. When the white smoke stopped evolving, an extra 100 mL dichloromethane, 10 mL triethylamine, and 10 mL water were added to the reaction mixture, resulting in the mixture changing color from purple to dark orange/brown. UV/Vis of an aliquot of the reaction mixture showed λmax = 868 nm, indicating conversion of SiCl_2_ to Si(OH)_2_. Solids were removed from the reaction mixture via filtration, and the organic phase was separated. The organic phase was dried with sodium sulfate, and the solids were removed by filtration. The solvent was removed in vacuo. The crude product was purified by CombiFlash chromatography using a 40 g pre-packed silica column with gradient elution, dichloromethane to 50% ethyl acetate in dichloromethane. The relevant fraction was collected, and the solvent was removed in vacuo, yielding 690 mg (58.5%) 4 as a brown crystalline solid. UV/Vis (toluene): λmax = 865 nm, ε = 2.91 × 10^5^ M^−1^ cm^−1^. A total of 1H NMR (400 MHz, C6D6) δ = 9.54 (m, 8H), 7.73 (m, 8H), 5.55 (m, 16H), 3.98 (m, 16H), 3.35 (s, 24H).

Silicon 5,9,14,18,23,27,32,36-octa(2′-methoxyethoxy)-2,3-naphthalocyanine di[(3″-mercaptopropyl)dimethylsiloxide] (9): A 100 mL 1 neck round bottom flask was charged with 690 mg (0.505 mmol) of 4 and a stir bar and fitted with a rubber septum. The reaction vessel was evacuated and refilled with argon three times, and the 60 mL of pyridine was injected, and the reaction vessel was evacuated and refilled with argon an additional three times. A total of 1.016 g (6.2 mmol) 3-mercaptropropylmethoxydimethylsilane was added to the reaction vessel via syringe transfer. The reaction vessel was immersed in a 130 °C oil bath and stirred for 16 h. The reaction vessel was cooled to approximately 80 °C and fitted with a short path distillation unit. Approximately half of the solvent was distilled at 145 °C through a 6-inch Vigreux column. The remaining pyridine was removed via vacuum distillation in a 60 °C oil bath with the receiving flask immersed in an ice bath. When the pyridine stopped distilling, the residual solvent was removed in vacuo with the reaction vessel in a 30 °C oil bath until the vacuum pressure was 0.75 torr. The reaction vessel was filled with argon, 75 mL of methanol was added, and the mixture was stirred and sonicated to disperse the solids. The mixture was centrifuged, and the supernatant was drained. Two additional methanol wash/centrifugation steps were performed, and remaining volatile components were removed in vacuo. A total of 780 mg 5 (94.7%) was collected as a brown crystalline powder. UV/Vis (toluene): λmax = 866 nm, ε = 3.40 × 10^5^ M^−1^ cm^−1^. A total of 1H NMR (400 MHz, C6D6) δ = 9.50 (m, 8H), 7.70 (m, 8H), 5.71 (m, 16H), 4.04 (m, 16H), 3.31 (s, 24H), 1.19 (q, J = 7.5 Hz, 4H), 1.12 (t, J = 7.0 Hz, 4H), −0.28 (m, 4H), −1.39 (m, 4H), −1.91 (s, 12H).

Silicon 5,9,14,18,23,27,32,36-octa(2′-methoxyethoxy) -2,3-naphthalocyanine di(3″-[([(N-succinimidyl)oxy]carbonyl)ethy]thiopropyldimethylsiloxide) (11): A 100 mL single neck round bottom flask was charged with a stir bar oven-dried. The reaction vessel was purged with argon, and then 780 mg (0.48 mmol) 5, 50 mL dichloromethane and 459 mg (2.7 mmol) N-succinimidyl acrylate, and 1.25 mL N,N-diisopropyl-N-ethyl amine. The headspace of the reaction vessel was purged with argon, and the reaction vessel was immersed in a 40 °C oil bath and stirred for 16 h. The reaction mixture was cooled, and the solvent was removed in vacuo while the reaction vessel was in a 40 °C water bath until the vacuum pressure was 0.5 torr. The 35 mL methanol was added to the crude product, the mixture was dispersed by sonication, and the solids were allowed to settle. The supernatant was removed, and 15 mL methanol was added to the crude product. The mixture was centrifuged, and the supernatant was drained. Residual methanol was removed in vacuo. The crude product was purified by CombiFlash chromatography on a 24 g pre-packed silica column with gradient elution, dichloromethane to 50% ethyl acetate in dichloromethane. The relevant fractions were collected, and the solvent was removed in vacuo, yielding 685 mg 6 (72.8%) as a brown crystalline powder. UV/Vis (toluene): λmax = 867 nm, ε = 3.27 × 10^5^ M^−1^ cm^−1^. A total of 1H NMR (500 MHz, C6D6) δ = 9.50 (m, 8H), 7.75 (m, 8H), 5.71 (m, 16H), 4.08 (t, J = 4.5 Hz, 16H), 3.36 (s, 24H), 1.99 (t, J = 7 Hz, 4H), 1.90 (t, J = 7.0 Hz, 4H), 1.54 (br. s, 4H), 1.42 (br. s, 4H), 1.30 (t, J = 7.5 Hz, 4H), 1.19 (q, J = 7.5 Hz, 4H), 1.12 (t, J = 7.0 Hz, 4H), −0.25 (m, 4H), −1.34 (m, 4H), −1.90 (s, 12H).

Silicon 5,9,14,18,23,27,32,36-octa(2′-methoxyethoxy)-2,3-naphthalocyanine mono[([([(N-succinimidyl)oxy]carbonyl)ethy]thio)propyldimethylsiloxide] mono( [([([bis(propyltrimethylammonium)amino]carbonyl)ethy]thio)propyl]dimethylsiloxide) dimesylate (4): A 20 mL scintillation vial was charged with 213 mg (0.108 mmol) 11, a stir bar, 6 mL chloroform and 24 µL (20.4 mg, 0.108 mmol) bis-(dimethylaminopropyl) amine. The reaction mixture was stirred 16 h at room temperature. A total of 0.1 mL (130 mg, 1.2 mmol) methyl mesylate was added to the reaction mixture, and the mixture stirred 24 h. The mixture was transferred to a 100 mL single neck round bottom flask, and the solvent was removed in vacuo. The crude product was washed with diethyl ether and dissolved into chloroform. The product was precipitated with diethyl ether and stirred in the flask. The supernatant was drained, and residual solvent was removed in vacuo, yielding 249 mg (quantitative yield) 1 as a brown waxy solid. UV/Vis: (solvent, λmax, ε): toluene, 869.9 nm, 1.39 × 10^5^ M^−1^ cm^−1^; ethanol, 858.1 nm, 2.02 × 10^5^ M^−1^ cm^−1^. A total of 1H NMR (CDCl3) δ = 9.10 (br. s, 8H), 7.94 (br. s, 8H), 5.36 (br. s, 16H), 4.10 (br. s, 16H), 3.53 (m, 24H), 2.73 (s, 4H), 2.56 (s, 24H), 2.16 (m, 4H), 2.05–1.99 (m, 4H), 1.31 (m, 4H), −0.29 (m, 2H), −0.45 (m, 2H), −1.47 (m, 2H), −1.61 (m, 2H), −2.26 (m, 12H).

### 4.2. UV-Visible Spectroscopy

The UV-Visible spectrum of the SiNC (1–4) dyes, ICG, and IRDye800 were recorded on an Evolution™ 300 UV-Vis Spectrophotometer (ThermoFisher Scientific, Waltham, MA, USA) by preparing solutions of the respective dyes in PBS or PBS with 0.1% BSA. Stocks of the different dyes were prepared in DMSO, and their respective concentrations were calculated using their extinction coefficients mentioned elsewhere in the manuscript. For ICG and IRDye800, the stocks were diluted in PBS, and the concentrations were calculated using their extinction coefficients 240,000 M^−1^ cm^−1^ and 113,790 M^−1^ cm^−1^, respectively.

### 4.3. Photobleaching Studies (Photostability)

Photobleaching effects of PA laser irradiation on the various naphthalocyanine dyes were studied using a custom-built PA imaging device that can monitor PA signal generated for every laser pulse [31,46]. Briefly, the fiber bundle of an OPO (Phocus HE Benchtop, OPOTEK, Carlsbad, CA, USA) was placed perpendicular to a single element transducer of 25 MHz central frequency (V324, Olympus, Waltham, MA, USA) that collected the generated PA signal. The vertical distance between the fiber and transducer was set to 1 inch to match the focus of the transducer. From the transducer, the signal was passed through a pulser/receiver (DPR500, Imaginant, Pittsford, NY, USA), which amplified the signal by 30 dB and then digitized using a data acquisition card (CSE161G2, GaGe, Poway, CA, USA). The pulser/receiver had 5 MHz high pass filter and a 150 MHz low pass filter to filter the generated PA signals. The dyes dissolved in Dimethyl sulfoxide (DMSO) were placed in polyethylene tubes (PE200, Intramedic, Franklin Lakes, NJ, USA) of 1.4 mm inner diameter and 1.9 mm outer diameter, which were subsequently mounted into a custom designed -3D printed box to be imaged at either 870 nm wavelength (NC dyes) or 800 nm wavelength (ICG and IRdye800CW). The PA signal change at the start of the experiment and at the end of 10 min irradiation was measured. The laser irradiation pulse repetition frequency was 10 Hz, and fluence was 30 mJ/cm^2^ (below ANSI standard for nanosecond pulsed laser; ANSI, American National Standard for Safe Use of Lasers, Laser Institute of America, 2014). The change in PA signal was computed as difference between the amplitude average of the first 20 pulses and amplitude average of 501–520 pulses.

### 4.4. Photoacoustic Measurements

Comparative PA studies of the different SiNC(1–4) dyes, IRDye800, and ICG were performed using a Vevo 2100 LAZR system (Visualsonics, FujiFilm, ON, Canada) equipped with a 21 MHz transducer (LZ250) and an Nd:YAG laser. Stock solutions (prepared in DMSO) of the different dyes were diluted in PBS at a concentration of 5 µM and loaded in polyethylene tubes (BTPE-50; Instech Laboratories Inc., Plymouth Meeting, PA, USA). Spectral PA measurements were performed in the wavelength range of 680 to 950 nm. PA signal intensity at the absorption maxima of the different dyes (mentioned elsewhere in the manuscript) was used to plot the PA signal intensity versus concentration graph post-adjustment of wavelength-dependent laser output energy. PA imaging parameters—PA gain, laser power, signal intensity, and persistence were kept constant across different measurements. The dye solutions placed inside a tube were also located at the same distance from the transducer to avoid variation in PA signal due to difference in fluence. Image analysis was performed using the built-in VevoLab 5.5.1 software.

### 4.5. Oral Squamous Cell Carcinoma Line (Cal 27) Culture

Human tongue squamous cell carcinoma line Cal27 was obtained from American Type Culture Collection (ATCC CRL-1624™, Manassas, VA, USA) and was cultured in DMEM Medium (10–013-CV, Corning, AZ, USA) supplemented with an antibiotic mixture containing Penicillin (100 I.U/mL) and streptomycin (100 μg/mL) (30–001-Cl, Corning, AZ, USA), and 10% heat-inactivated fetal bovine serum (FBS) (SH30071.03HI, Hyclone™, Marlborough, MA, USA). Cells used in this study tested negative for Mycoplasma contamination via MycoAlert™ PLUS Mycoplasma Detection Kit (Lonza, Basel, Switzerland).

### 4.6. Preparation of Tumor Cell Phantom Gelatin Molds for PA Imaging

Tumor cell phantom preparation and imaging were performed as described in our previously published report [25]. Briefly, cells were seeded in 100 mm petri dishes and cultured for 24 h. Subsequently, 2.5 µM of the different dyes were added, and the cells were further incubated. After the desired incubation period (2 h or 24 h), the cells were washed, collected by gently scraping by a cell scraper, counted, and 0.25 × 10^6^ cells were resuspended in 5–8% gelatin. In order to generate a tissue mimicking model, the resuspended cells were immediately poured into the wells of a 96-well plate previously prepared to have a layer of 10% gelatin (100 μL) to minimize reverberations from the base of the 96-well plate during PA imaging [25,47]. The gelatin molds were allowed to solidify, and the plates were imaged within 24 h. PA imaging was performed by placing the 96-well plate in water. PA imaging parameters such as the PA gain, laser power, signal intensity, persistence, and frame rate were kept constant across different measurements. Image analysis was performed using the built-in software provided by Visualsonics, FujiFilm (Toronto, ON, Canada).

## 5. Conclusions

To summarize, in this study, we report the synthesis and characterization of SiNC dyes with different water solubilities and electronic properties. While similar dyes have been reported in the past, we demonstrate the possibility of incorporating a functional group for further bioconjugation of these dyes. SiNC dyes have been reported previously; however, their absorption profiles have been restricted to the 800 nm region. In this study, we demonstrate the synthesis of two water-soluble SiNC dyes; SiNC-3 and SiNC-4, with absorption maxima around 900 nm, which may be ideal for PA imaging as it is away from the hemoglobin absorption window. The absorption spectrum of SiNC-3 was red-shifted as compared to other SiNC dyes, and it showed a higher PA contrast with a higher cellular PA signal highlighting its utility as a PA contrast agent. While the photostability of SiNC-3 was comparable to that of IRDye800 and ICG, it was significantly lower than that of SiNC-4. To conclude, out of the four SiNC dyes synthesized in our study, SiNC-3 and SiNC-4 were found to be the best-performing dyes in terms of water solubility, PA contrast, and cellular PA imaging. Future studies will be focused on validating the biocompatibility and bioconjugation of these dyes and demonstrating their ability for targeted tumor imaging.

## Data Availability

Data reported in the study are available from the corresponding author on reasonable request.

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
