# Peer review of "Optimizing Axial and Peripheral Substitutions in Si-Centered Naphthalocyanine Dyes for Enhancing Aqueous Solubility and Photoacoustic Signal Intensity"

_ijms, 2023, doi:10.3390/ijms24032241_

Round 1

Reviewer 1 Report

NIR absorption dyes and nanoparticles with photostability and aqueous solubility are very useful for optical imaging and thermotherapy in deep human tissue. In this manuscript, the authors developed  SiNC dyes with good water solubilities and low photobleaching, and demonstrated the synthesis of 2 dyes absorption maxima beyond 900 nm which may be ideal for PA imaging as it is away from the hemoglobin absorption window. These dyes have the potential of biocompatibility and bioconjugation, which probably have the ability for targeted tumor imaging in deep tissue.

Small drawbacks:

Line 221, “2.5 * 106” should be “2.5*10^6”.

Line 579, “240000 M-1cm-1 and 113790 M-1cm-1”, should be “240000 M^-1cm^-1 and 113790 M^-1cm^-1”.

Author Response

We thank the reviewer for the comment. Typographical errors and inconsistencies in the units have been corrected. We have also thoroughly read the manuscript and edited it for language and grammar.

Reviewer 2 Report

     The manuscript focuses on the improvement of previously synthesized photoacoustic contrast agents for more practical use. The research topic is very relevant in the field of the photoacostic imaging (PAI) modality, although the authors had published a reserch article about silicon-centered naphthalocyanime dyes in the Referene 24. However, the content is a lack of organized description, although the experimental design is reasonable. I have questions and other suggestions below. Overall, I only recommend this manuscript for publication after major revisions; however, if the publisher decides to reject the manuscript, it is also acceptable: 

  1. Introduction. There is a good starting point in Line 47—51 to describe the principle of the photoacostic imaging (PAI) modality. It is better to explore what types of PAI contrast agents, how to design or choose contrast agents.  PAI contrast agents are limited not only to organic dyes. In Line 51–58, it is to tendentiously describe the other imaging modalities; all modalities have their advantages and limitations; PAI currently cannot fully replace X-ray, CT, and PET, at least due to imaging depth. In terms of the resolution comparison between PAI and MRI, it depends on the configuration of the imaging system. (1) 

  1. Please consider to thicken chemical bonds in Figure 1 and Scheme 1. 

  1. Keep the passive tone in the writing in Line 105124, 159, 169–171, 220–223  

  1. Please make the format consistent. For example, in Line 98, 133, 140, 151, 160, 239 and 665, the f in figure should be capitalized. In Line 239, there should be a space between figure and 1. In Line 253, please correct the comma location. 

  1. In Line 137–138, 169, please clarify the sentence. 

  1. In Line 140, please check Reference 28. The reference does not match the description. 

  1. In Line 141–142, please revise “100% aqueous solvents” as water. 

  1. In Line 142–143, which curve of the ratio do you refer to? Please provide the ratio vaules.  

  1. In Figure 2, Please label SiNC1–4 in the sub-pannels (same as the layout in Figure 3); provide the full name of DMAC, and more detailed in the figure description, especially for the Water curve in sub-pannel D. 

  1. In Figure 4, could you provide more information on the setup of the sample (similar to Figure 2 in Reference 24)? The setup information could be added to the supplement. Calibration bar or signal intensity bar? In Figure 4B, the PA images show a single red line in the top 3 panels, and two red lines in the bottom 3 panels; how to we determine the intensity of the PA signal? 

  1. In Line 206–207, please revise the sentence as NC dyes in DMSO. 

  1. Please assign A and B letters in Figure 6. 

  1. In Line 225, why do you take Cal 27 cells as an in vitro cell model? The reason could be addressed in the Introduction.  

  1. Line 276–279 could be moved before Line 251. 

  1. Please carefully check the format in Section 4 (Materials and Methods): the space format and the unit of extinction coefficients (Line 579). 

  1. Summary (Line 333341) and Conclusion (Line 638–647) could be combined to reduce the redundant information. 

  1. Please provide a systematic and logical summary or conclusion about which number of SiNC is the best contrast agent. No clear final answer to show which is the best SiNC, based on the mentioned information: in Line 136, SiNC-3 and -4 aggregation was observed in water. In Line 185, the PA signal intensity of SiNC1-3 was higher than that of SiNC-4. In Line 211–212, SiNC-4 was claimed as an apt photoacoustic imaging contrast agent based on photostability. 

  1. In Line 663, Appendix A could be removed. All the information was shown in the supplementary. 

  1. Supplementary figure 1 (Line 239) should be revised as Figure 1S. 

Reference 

1. B. Shrestha, F. DeLuna, M. A. Anastasio, J. Y. Ye, E. M. Brey, “Photoacoustic Imaging in Tissue Engineering and Regenerative Medicine,” Tissue Eng Part B Rev, 26(1):79-102 (2020).

Author Response

The manuscript focuses on the improvement of previously synthesized photoacoustic contrast agents for more practical use. The research topic is very relevant in the field of the photoacoustic imaging (PAI) modality, although the authors had published a research article about silicon-centered naphthalocyanine dyes in the Reference 24. However, the content is a lack of organized description, although the experimental design is reasonable. I have questions and other suggestions below. Overall, I only recommend this manuscript for publication after major revisions; however, if the publisher decides to reject the manuscript, it is also acceptable: 

We take the reviewer’s criticism positively and have tried to incorporate all the suggestions. The reviewer rightly points out at the previously published manuscript (reference 24) by our group, which describes the synthesis, bioconjugation and application of a hydrophobic (water insoluble) SiNC dye. Since low aqueous solubility limits the bioconjugation of these dyes to antibodies and other targeting moieties, the objective of the current study was to chemically alter the SiNC dye structure (previously reported in references 23 and 24) to generate water soluble SiNC dyes, while preserving their absorption profiles and photoacoustic properties. We do apologize for the “lack of organized description” and have extensively modified the manuscript based on the reviewers’ suggestions and further edits/reshuffling of text to make it more organized.   

  1. There is a good starting point in Line 47—51 to describe the principle of the photoacoustic imaging (PAI) modality. It is better to explore what types of PAI contrast agents, how to design or choose contrast agents.  PAI contrast agents are limited not only to organic dyes. In Line 51–58, it is to tendentiously describe the other imaging modalities; all modalities have their advantages and limitations; PAI currently cannot fully replace X-ray, CT, and PET, at least due to imaging depth. In terms of the resolution comparison between PAI and MRI, it depends on the configuration of the imaging system. (1)

We do agree with the reviewer’s suggestion and have reshuffled the sentences in the introduction to bring up the principle of photoacoustic imaging in the beginning. Also, included later on in the introduction (2nd paragraph) is a brief description of several photoacoustic contrast agents and their limitations, which sets the tone for our study and why the photoacoustic contrast agents designed in this study may have advantages over the several others that are reported. The statements describing the comparison of photoacoustic imaging with other imaging modalities have been modified to reflect the advantages and shortcomings of those techniques and why photoacoustic imaging may be an important tool for some specific applications. The reference suggested by the reviewer has also been cited.

  1. Please consider to thicken chemical bonds in Figure 1 and Scheme 1.

We have added a high-resolution image of the same.

  1. Keep the passive tone in the writing in Line 105–124, 159, 169–171, 220–223 

             This has been changed as per suggestions.

  1. Please make the format consistent. For example, in Line 98, 133, 140, 151, 160, 239 and 665, the f in figure should be capitalized. In Line 239, there should be a space between figure and 1. In Line 253, please correct the comma location.

             This has been changed as per suggestions.

  1. In Line 137–138, 169, please clarify the sentence.

We apologize for the ambiguity in the sentences and have now modified them extensively.

  1. In Line 140, please check Reference 28. The reference does not match the description.

We do agree that reference 28 is not directly related to the J aggregation of NC dyes that is being discussed in line 140. However, this reference was included to provide support for the statement that red-shift in spectra is a characteristic of J-aggregation. 

  1. In Line 141–142, please revise “100% aqueous solvents” as water. 

We apologize for this confusion and have now modified this to “aqueous solvents”.

  1. In Line 142–143, which curve of the ratio do you refer to? Please provide the ratio vaules. 

             We apologize if this was not mentioned clearly. The ratio in that is discussed in lines 142-143 is the absorbance of the Q-band (at 860 – 880 nm) to the vibrational shoulder (at 750 – 800 nm). We have made this clear in the revised manuscript and added the ratio values.

  1. In Figure 2, Please label SiNC1–4 in the sub-pannels (same as the layout in Figure 3); provide the full name of DMAC, and more detailed in the figure description, especially for the Water curve in sub-pannel D.

We thank the reviewer for pointing this out and apologize for not providing the necessary details in the figure legends. The details have now been added.

  1. In Figure 4, could you provide more information on the setup of the sample (similar to Figure 2 in Reference 24)? The setup information could be added to the supplement. Calibration bar or signal intensity bar? In Figure 4B, the PA images show a single red line in the top 3 panels, and two red lines in the bottom 3 panels; how to we determine the intensity of the PA signal? 

We apologize for not clearly stating the experimental setup and description. We have now included an experimental schematic (supplementary figure S1). To further clarify the experimental setup, the orientation of images in figure 4B have been labeled. The calibration/signal intensity bar has been made more clear.

The two red lines observed for the water-soluble samples in the bottom panel represent the top and bottom of the tube (holding the sample). However, for the restively insoluble samples (upper panel) the single red line represents the bottom of the tube where the aggregated sample tends to settle over time. It is to be noted that all samples were placed at an equal distance from the transducer. While the intensities for the two lines observed in the water-soluble samples (lower panel) were similar, for consistency in quantification method of the water soluble and insoluble samples, intensity at the lower line was used for quantification.

  1. In Line 206–207, please revise the sentence as NC dyes in DMSO.

This statement has been added.

  1. Please assign A and B letters in Figure 6.

             Figures 6A and 6B have been labeled as per suggestion.

  1. In Line 225, why do you take Cal 27 cells as an in vitro cell model? The reason could be addressed in the Introduction. 

             This statement has been added.

             “The human tongue squamous cell carcinoma line (Cal 27) was used to test the uptake and                  imaging efficacy of the dyes in an in vitro oral cancer model.” 

  1. Line 276–279 could be moved before Line 251.

As suggested by the reviewer the lines 276-279 have been moved before line 251.

  1. Please carefully check the format in Section 4 (Materials and Methods): the space format and the unit of extinction coefficients (Line 579). 

This has been checked and edited as per suggestions.

  1. Summary (Line 333–341) and Conclusion (Line 638–647) could be combined to reduce the redundant information. 

We do agree with the reviewer and have deleted the summary line 333-341 to avoid redundance.

  1. Please provide a systematic and logical summary or conclusion about which number of SiNC is the best contrast agent. No clear final answer to show which is the best SiNC, based on the mentioned information: in Line 136, SiNC-3 and -4 aggregation was observed in water. In Line 185, the PA signal intensity of SiNC1-3 was higher than that of SiNC-4. In Line 211–212, SiNC-4 was claimed as an apt photoacoustic imaging contrast agent based on photostability. 

We thank the reviewer for this important comment, and we have now modified the conclusion to incorporate the summary and outcome of the different SiNC dye performance and suggest which one could be the best for further exploration.

  1. In Line 663, Appendix A could be removed. All the information was shown in the supplementary. 

We do agree with the reviewer and have removed Appendix A.

  1. Supplementary figure 1 (Line 239) should be revised as Figure 1S. 

This has been revised as per suggestions.

Round 2

Reviewer 2 Report

     The manuscript has been improved a bit after revision. However, the authors should pay more attention, especially since the same mistakes happened in the second revision. I still suggest some minor revisions before accepting this manuscript for publication:

1.     Introduction. In Line 55-62, clarifying the sentences is required. Referencing 1 could be an example.

2.     In Line 120-139, the tone in sentences did not revise as passive tone yet (please carefully go through the whole manuscript). Furthermore, for the conjugation of function groups on the molecules, 'install' is not a professional word suggested for use in the articles.

3.     In Table 1, the format is messy.

4.     In Line 169, please also include at least one reference directly mentioned J aggregation.

5.     In Line 172-175, what does the ratio refer to? Normalized absorbance or molar extinction coefficient? Focus on blue curves? In Line 174, the ratio is ~4; in Line 175, the ratio is ~2.5. If the decimal place was kept to show ~2.5, the ratio should be around 3.3 for SiNC-2 and SiNC-3.

6.     In Figure 2 and Figure 6, the right parentheses assigned to the subpanels are not required. In Line 182, the parentheses next to D are not required.

7.     In Line 233-234, the calibration/intensity bar is shown, but in Figure 4 the 'Photoacoustic signal intensity (au)' is noted above the bar. PA is used as the abstraction of photoacoustic; why not write the PA signal intensity? The meaning to describe the bar is so confusing: calibration/intensity bar, PA signal intensity, or calibrated PA signal intensity. Usually, the names of the bars in the figures and descriptions should be the same. However, if there is not enough space in the figures, the more detailed information (calibrated PA signal intensity (a.u.)) should be shown in the legend, and the scale intensity bar should just be noted as 'intensity' or just a bar with Min and Max labeling.

8.     In the section of the Discussion (Line 308), there is no space at the beginning of paragraphs.

9.     In Line 708, it is better to use two instead of 2.

10.  In Supplementary, capitalize the f in figure S#.

Reference 1. M. A Attawia, J. E. Devin, C. T. Laurencin, “Immunofluorescence and confocal laser scanning microscopy studies of osteoblast growth and phenotypic expression in three-dimensional degradable synthetic matrices,” J Biomed Mater Res, 29, 843-8 (1995).

Author Response

We thank the reviewer for reviewing the manuscript and providing some really insightful comments and feedback. The point-by-point response comprehensively addresses each comment.

The manuscript has been improved a bit after revision. However, the authors should pay more attention, especially since the same mistakes happened in the second revision. I still suggest some minor revisions before accepting this manuscript for publication:

  1. Introduction. In Line 55-62, clarifying the sentences is required. Referencing 1 could be an example.

We thank the reviewer for suggesting this change. The paragraph has been thoroughly revised as suggested using reference 1 as an example.

  1. In Line 120-139, the tone in sentences did not revise as passive tone yet (please carefully go through the whole manuscript). Furthermore, for the conjugation of function groups on the molecules, 'install' is not a professional word suggested for use in the articles.

We apologize for the tone of these sentences and the use of the scientifically inappropriate term “install”. The suggested lines have been modified extensively to address the suggestions. Also, the entire manuscript has been read thoroughly to maintain consistency in tone.

  1. In Table 1, the format is messy.

The format of table 1 appears messy due to the page formatting. We have modified it to make it more clear.

  1. In Line 169, please also include at least one reference directly mentioned J aggregation.

As suggested, two references have been added to support the statement.

K.; Duan, X.; Jiang, Z.; Ding, D.; Chen, Y.; Zhang, G.-Q.; Liu, Z. J-aggregates of meso-[2.2]paracyclophanyl-BODIPY dye for NIR-II imaging. Nature Communications 2021, 12, 2376.

Würthner, F.; Kaiser, T.E.; Saha-Möller, C.R. J-Aggregates: From Serendipitous Discovery to Supramolecular Engi-neering of Functional Dye Materials. Angew. Chem. Int. Ed., 2011, 50, 3376-3410.

  1. In Line 172-175, what does the ratio refer to? Normalized absorbance or molar extinction coefficient? Focus on blue curves? In Line 174, the ratio is ~4; in Line 175, the ratio is ~2.5. If the decimal place was kept to show ~2.5, the ratio should be around 3.3 for SiNC-2 and SiNC-3.

We apologize for this confusion and thank the reviewer for pointing this out. We have now modified this section and made it clear. The ratios are calculated based on the normalized absorbance (blue curves plotted on the secondary y-axis).

  1. In Figure 2 and Figure 6, the right parentheses assigned to the subpanels are not required. In Line 182, the parentheses next to D are not required.

As suggested, we have now removed the parentheses next to D in  line 182. We have retained the parentheses in figure 2 and 6 to make it consistent with the rest of the figures which also have parentheses.

  1. In Line 233-234, the calibration/intensity bar is shown, but in Figure 4 the 'Photoacoustic signal intensity (au)' is noted above the bar. PA is used as the abstraction of photoacoustic; why not write the PA signal intensity? The meaning to describe the bar is so confusing: calibration/intensity bar, PA signal intensity, or calibrated PA signal intensity. Usually, the names of the bars in the figures and descriptions should be the same. However, if there is not enough space in the figures, the more detailed information (calibrated PA signal intensity (a.u.)) should be shown in the legend, and the scale intensity bar should just be noted as 'intensity' or just a bar with Min and Max labeling.

We apologize for this confusion and have modified the figure4 legend to match with what is shown in the figure. We have made similar changes for figure 6 and supplementary figure 2 as well to maintain consistency.

  1. In the section of the Discussion (Line 308), there is no space at the beginning of paragraphs.

Space has been added at the beginning of the paragraphs, as suggested.

  1. In Line 708, it is better to use two instead of 2.

This has been modified as per suggestion.

  1. In Supplementary, capitalize the f in figure S#.

This has been modified as per suggestion.

Reference 1. M. A Attawia, J. E. Devin, C. T. Laurencin, “Immunofluorescence and confocal laser scanning microscopy studies of osteoblast growth and phenotypic expression in three-dimensional degradable synthetic matrices,” J Biomed Mater Res, 29, 843-8 (1995).
